# Effects of Adherence to Pharmacological Treatment on the Recovery of Patients with Schizophrenia

**DOI:** 10.3390/healthcare9091230

**Published:** 2021-09-18

**Authors:** Alejandra Caqueo-Urízar, Alfonso Urzúa, Patricio Mena-Chamorro, Josefa Bravo de la Fuente

**Affiliations:** 1Instituto de Alta Investigación, Universidad de Tarapacá, Arica 1001236, Chile; 2Escuela de Psicología, Universidad Católica del Norte, Antofagasta 1270709, Chile; alurzua@ucn.cl; 3Departamento de Psicología, Universidad de la Frontera, Temuco 4811322, Chile; pmena@uta.cl; 4Escuela de Psicología y Filosofía, Universidad de Tarapacá, Arica 1010069, Chile; josefar.b.dlf@gmail.com

**Keywords:** medication adherence, personal recovery, schizophrenia

## Abstract

The aim of this study was to evaluate the effects of adherence to antipsychotic treatment on the recovery of patients with schizophrenia in northern Chile. One hundred and fifty-one patients diagnosed with schizophrenia completed the Drug Attitude Inventory (DAI-10), Positive and Negative Syndrome Scale for Schizophrenia (PANSS), Recovery Assessment Scale (RAS-24), sociodemographic information, and clinical and treatment characteristics of patients with schizophrenia. Multivariate analysis with multiple linear regression was then performed to identify variables that were potentially associated with the recovery assessment (variable criterion). A significant association was found between adherence to antipsychotic medication and the Willing to Ask for Help dimension of Recovery (β = 0.239, *p* = 0.005). Association of clinical and socio-demographic variables with recovery were identified: negative symptoms with Personal Confidence and Hope (β = −0.341, *p* = 0.001) and Goal and Success Orientation (β = −0.266, *p* = 0.014); cognitive symptoms with Willing to Ask for Help (β = −0.305, *p* = 0.018) and no domination by symptoms (β = −0.351, *p* = 0.005); marital status with reliance on others (β = −0.181, *p* = 0.045); age with Personal Confidence and Hope (β = −0.217, *p* = 0.021), Goal and Success Orientation (β = −0.296, *p* = 0.003), and no domination by symptoms (β = 0.214, *p* = 0.025). Adherence has a positive relationship with personal recovery in this sample of Chilean patients with schizophrenia.

## 1. Introduction

The lack of adherence to treatment is one of the most important health problems and has come to be called the “invisible epidemic” [1,2,3]. The World Health Organization (WHO) considers adherence to treatment as taking medication according to the prescribed dosage and with persistence over time [4].

Difficulties with treatment adherence in patients with schizophrenia can lead to relapses, which on average can be five times more likely in young men during their first psychotic episode than in women, increasing the likelihood of hospitalization and healthcare costs [5,6,7,8,9,10].

Five areas were observed that were configured as factors influencing adherence: personal, systemic, disease-related, sociodemographic, and treatment-related factors. The first involves the patients’ perception of the efficacy of the medication, adverse effects, and certain types of personal or religious beliefs that influence their view of consuming medications [11]. Systemic factors are related to the provision of care by different health professionals and polypharmacy; third, there is the presence of comorbidity, among which substance abuse, high levels of hostility, and cognitive failures stand out. Fourth, there are sociodemographic factors and finally, factors related to treatment are oriented toward the duration of treatment and the frequency and intensity of side effects [10,12,13,14].

Other aspects should also be considered, such as premorbid, sociodemographic, socioeconomic, and cultural factors. Studies have indicated that a lower level of adherence to pharmacological treatment has been observed in patients belonging to an ethnic minority or with a lower educational level [9,13,14,15,16,17,18,19,20,21,22,23,24,25].

Adherence is associated with recovery, which has been defined both as a process that incorporates clinical components, such as symptom remission and functionality [26,27,28,29] as well as subjective components associated with the development of individualized coping mechanisms and improvements in levels of psychological well-being [30,31]. These components reflect different perspectives not necessarily accordant with each other and usually encompass both the patient’s and the healthcare team’s point of view [29,32,33]. Studies have shown that a lack of adherence negatively influences patient recovery by increasing the likelihood of relapse [34,35,36,37]. 

Although studies have shown the associations noted above, most of these investigations have been conducted in contexts other than Latin America, thus, it is of consequence to establish evidence presenting a different socioeconomic and cultural context. The objective of this study was to evaluate the effects of treatment adherence on the recovery of patients with schizophrenia in northern Chile.

The following scales were used. Drug Attitude Inventory (DAI-10) [38]: This 10-item self-report scale was developed to assess attitudes, experiences, and beliefs about antipsychotic drugs. Recovery Assessment Scale (RAS-24) [39]: This scale evaluates the subjective assessment of recovery through 24 items that have resulted from the factor analysis of the original scale consisting of 41 items. Positive and Negative Syndrome Scale for Schizophrenia (PANSS) [40]: This 30-item self-report scale was developed to assess psychotic symptoms in individuals with schizophrenia.

## 2. Materials and Methods

### 2.1. Study Participants

This cross-sectional study analyzed the information obtained through interviews and surveys of patients diagnosed with schizophrenia at the Public Mental Health Centers in the city of Arica, Chile. Patients were invited to participate as they came for their monthly follow-up visits. A set of exclusion criteria for the selection of patients was applied (being in a state of psychotic crisis or having a sensory or cognitive type of disorder that prevents being evaluated) to ensure their ability to participate fully in the interviews. As most of the patients were stable, in relation to their psychotic symptoms, the number of patients excluded was low, and the majority of the participants agreed to participate.

The final sample included 151 patients with an ICD-10 diagnosis of Schizophrenia [41].

### 2.2. Measures

Drug Attitude Inventory (DAI-10) [38]: This 10-item self-report scale was developed to assess attitudes, experiences, and beliefs about antipsychotic drugs. The DAI-10 is considered a good predictor of adherence to treatment in schizophrenia [38,42]. The response options are in true-false form (1 = “true” to −1 = “false”). Scores ranged from −10 (very poor attitude) to +10 (best possible attitude). Patients with a score of six to 10 were considered adherent, from zero to five, moderate, and in negative ranges were considered non-adherent [43]. The Spanish version of this instrument was developed by Ramírez et al. in 2004 [44]. The instrument’s psychometric properties showed an inter-rater reliability index of 0.61 (*p* < 0.001) and an internal consistency coefficient of 0.57. The version of the DAI has convergent validity as well as moderate reliability [44]. This is a simple and easy-to-use self-report instrument with good psychometric properties that assess a unique clinical dimension relevant to non-adherence [42]. The DAI-10 scores analyzed in this study were obtained from patients.

Recovery Assessment Scale (RAS-24) [39]: This scale evaluates the subjective assessment of recovery through 24 items that have resulted from the factor analysis of the original scale consisting of 41 items. The factors that make up the scale are personal confidence and hope (PCH, nine items), willingness to ask for help (WAH, three items), goal and success orientation (GSO, five items), reliance on others (RO, four items), and no domination by symptoms (NDS, three items). The response options are on a five-level Likert format (1 = “Strongly disagree” to 5 = “Strongly agree”). Currently, there is no cut-off point for interpreting RAS-24 scores; thus, to reduce arbitrariness, the scores were interpreted using quartiles (Q1 = 3.29, Q2 = 3.75, Q3 = 4.21). High scores suggest a more advanced recovery process. The RAS-21 presents adequate evidence of reliability and validity [39], has been translated in Spain by Muñoz et al. (2011), and Zalazar et al. (2017) examined the psychometric properties of this instrument in Argentina [45,46].

Positive and Negative Syndrome Scale for Schizophrenia (PANSS) [40]: This 30-item self-report scale was developed to assess psychotic symptoms in individuals with schizophrenia. For the purposes of this study, we considered five subscales of the PANSS: positive (five items), negative (seven items), excitation (five items), depression (four items), and cognitive (three items) symptoms [47]. The response options were on a seven-level Likert format (1 = “absent” to 7 = “extreme”). The scores to be interpreted were obtained by calculating the sum of all responses; the cut-off points were of Leucht et al. (2005) [48], where a PANSS total score of 58 suggests “mildly ill,” a PANSS of 75 to “moderately ill,” a PANSS of 95 to “markedly ill” and a PANSS of 116 to “severely ill.” The PANSS has been translated and validated in Spain by Peralta and Cuesta (1994), and Fresán et al. (2005) examined the psychometric properties of this instrument in Mexico [49,50].

Clinical and Demographic data: Age at onset of the disorder (defined as the age at which the first acute psychotic episode appears) and age at onset of treatment were included as clinical variables. The demographic variables assessed were gender, age, marital status (single or in couple), educational level (≥12 years or <12), employment status (unemployed or employed), ethnicity (Aymara and non-Aymara), and family income (measure of the total salary per month for all members of the family, expressed in US dollars).

Concerning ethnicity, the Aymara is the largest ethnic group, which has lived in the Andes Mountains for centuries, with a total population of 2 million people in Latin America. Recent generations of Aymara have undertaken a massive migration from rural towns to large cities and, thus, receive healthcare services from the same clinics as non-Aymara individuals [51,52,53,54].

### 2.3. Procedures

The study was approved by the Ethics Committee of the University of Tarapacá and the National Health Service of Chile. Two psychologists, who were part of the research team and supervised by the principal researcher, conducted the survey of the patients under the auspices of the mental health services. The duration of the evaluation (interview) was between 20 and 30 min. 

Written informed consent was obtained from the patients prior to the start of the survey. The objectives of the study were explained, as well as the voluntary nature of participation. No compensation was offered for participation in the study. For each center, during a three-month window, all patients were invited to participate as they came for their monthly follow-up visits. The majority of the patients agreed to participate.

### 2.4. Statistical Analysis

Initially, to characterize the sample, the proportions of each categorical variable were obtained, and the mean, standard deviation, minimum and maximum, skewness, kurtosis, and Shapiro-Wilk normality test [55] were calculated for each continuous variable. Comparisons were performed using the t-test for independent samples, based on the mean scores of each dimension of the recovery variable as a function of the sociodemographic variables (gender, marital status, ethnicity, religion, occupation, educational level). The dimensions of the recovery variable did not present homoscedasticity in marital status (WAH: Levene’s test, F_(1)_ = 4.436, *p* = 0.037; RO: Levene’s test, F_(1)_ = 7.864, *p* = 0.006); therefore, Welch’s *t*-test was used for this variable. The effect size of the differences was estimated using the coefficient d proposed by Cohen (1988) [56]. Although the quantitative variables in this sample are not normally distributed, parametric comparative analyses were used because the t-statistic is sufficiently robust under conditions of skewness and with large sample sizes (n > 60) [57,58].

The association among recovery variable dimensions (i.e., RAS-24), adherence to antipsychotic treatment (i.e., DAI-10), and psychotic symptom severity dimensions (i.e., PANSS) were estimated using Pearson’s correlation coefficient. Subsequently, to assess the potential predictive ability of adherence on the recovery of patients with a diagnosis of schizophrenia (criterion variable), a multiple linear regression analysis was performed. The dimensions of the RAS were considered separate-dependent variables (i.e., HCP, WAH, GSO, RO, and NDS). In addition, in an effort to control the influence of sociodemographic and clinical variables in the regression models, gender, marital status, age, ethnicity, religion, occupation, educational level, and severity of psychotic symptoms (i.e., PANSS positive, PANSS negative, PANSS cognitive, PANSS depression, and PANSS excitement), age of illness onset, and treatment were included. The final model incorporated standardized beta coefficients, which represent the changes in the standard deviation of the criterion variable. The predictor variables with the largest standardized beta coefficients suggest a greater relative effect on the recovery of patients diagnosed with schizophrenia. All assumptions were met. The presence of multicollinearity among the independent variables was ruled out by the tolerance level and inflated variance factor (IVF), which was greater than 0.1 and less than 10 for all, respectively. Residuals were independent of each other (PCH: Durbin-Watson = 1.975, WAH: Durbin-Watson = 1.902, GSO: Durbin-Watson = 1.992, RO: Durbin-Watson = 2.181, NDS: Durbin-Watson = 1.963). Homoscedasticity was confirmed using a scatter plot of predictors and standardized residuals. The normality of the residuals for each dependent variable was tested using a histogram and Q-Q plot of the standardized residuals. 

Statistical hypothesis testing of the data analyses was performed at a 5% significance level. All statistical analyses were performed using IBM SPSS version 25 software [59] and JASP version 0.14.1 [60].

## 3. Results

One hundred and fifty-one stabilized patients diagnosed with schizophrenia participated in this study. The mean age was 39.9 years (SD = 14.5), 94 (62.3%) were male, 121 (80.2%) had no partner, 136 (90%) had less than 12 years of education, 128 (84.7%) were unemployed, 67 (44.3%) self-declared Aymara, and the mean monthly family salary was $216.4 (SD = 206.3). Overall, the age at presentation of the first acute psychotic episode was 21.4 years (SD = 8.4) and that of treatment initiation was 23.8 years (SD = 8.9). All patients received pharmacological treatment with antipsychotics. According to adherence, 15.2% of patients diagnosed with schizophrenia were considered nonadherent. Only 4.6% reported severe psychotic symptoms and 25% presented mean scores above the 75th percentile (Q3 = 4.21), suggesting that most responses focused on reporting a high degree of agreement with a more advanced recovery process (min–max: 1–5). Sociodemographic details are presented in Table 1.

Patients with a diagnosis of schizophrenia showed moderate adherence to antipsychotic treatment (M = 3.7, SD = 0.7) and mean scores close to the 50th percentile (Q2 = 3. 75) on the RAS-24 dimensions, suggesting that most responses focused on reporting moderate agreement with statements about hope for the future, self-confidence, seeking help from others, desires to succeed, importance of others in recovery, and psychiatric symptoms no longer being the focus of personal life. Patients also had scores near the bottom of each dimension of the PANSS, suggesting positive, negative, cognitive, depressive, and mild excitatory symptoms. According to the standardized skewness and kurtosis coefficients, it can be observed that in all dimensions, only the skewness values were outside the recommended range (−2 to 2) [58]. This suggests that the variables had a positively or negatively skewed meso-kurtosis distribution. The Shapiro-Wilk test showed that none of the variables had a normal distribution. Descriptive details of the quantitative variables are presented in Table 2.

The t-test for independent samples showed statistically significant differences in the personal confidence and hope PCH dimensions according to educational level (>12 years: M = 4.08 [SD = 0.86]; <12 years: M = 3. 57 (SD = 0.77); *t*-Student = 2.341; *p* = 0.021; d = 0.657); in the Willing to Ask for Help (WAH) dimension according to marital status (without a partner: M = 3.70 [SD = 1.1]; with a partner: M = 4.36 [SD = 0.81]; t-Welch = 3.607; *p* ≤ 0.001; d = 0.649); On the Goal and Success Orientation (GSO) dimension by marital status (no partner: M = 3.70 [SD = 1.0]; partner: M = 4.17 [SD = 0.91]; t-Student = 2.270; *p* = 0.025; d = 0.463) and educational level (>12 years: M = 4.37 [SD = 0.64]; <12 years: M = 3.73 [SD = 1.0]; *t*-Student = 2.220; *p* = 0.028; d = 0.623), and reliance on other (RO) dimensions according to marital status (No partner: M = 3.72 [SD = 0.91]; partner: M = 4.28 [SD = 0.54]; *t*-Welch = 4.320; *p* < 0.001; d = 0.746). 

In general, these findings show that patients with more years of education could have a higher level of agreement with hope for the future and desire to succeed; additionally, those who are in a relationship could achieve a higher degree of agreement with the importance of others in recovery, the fulfillment of goals, and seeking help from others. 

It should be considered that the differences obtained were characterized by small to moderate effect sizes [56]. In addition, no statistically significant differences were found between the dimensions of the RAS-24 according to gender, ethnicity, and religion. 

Pearson’s correlation analyses showed that the recovery dimensions (RAS-24) had mostly mild and statistically significant direct correlations with adherence to antipsychotic treatment (DAI-10), as well as mild to moderate inverse correlations other than 0 in the population with the dimensions of psychotic symptom severity (PANSS) (see Table 2).

In multiple linear regression models, adherence to antipsychotic treatment had only a slight, statistically significant direct effect on the Willing to Ask for Help (WAH) dimension (β = 0.239; *p* = 0.005), suggesting that patients with higher adherence will be more interested in seeking help from others. Among the clinical variables, negative psychotic symptom severity (negative PANSS) had mild, inverse, statistically significant effects on personal confidence and hope (PCH) dimension (β = 0.341; *p* = 0.001) and Goal and Success Orientation (GSO) (β = −0.266; *p* = 0.014), while cognitive symptom severity (cognitive PANSS) had moderate and inverse effects, different from 0 on Willing to Ask for Help (WAH) (β = −0.305; *p* = 0.018) and no dominance by symptoms (NDS) (β = −0.351; *p* = 0.005), implying that the greater the severity of negative psychotic and cognitive symptoms, the lower the degree of recovery. According to sociodemographic variables, marital status had a mild inverse effect on Reliance on others (RO) (β = −0.181; *p* = 0.045) and age had mild inverse effects on personal confidence and hope for PCH (β = −0.217; *p* = 0.021), Goal and Success Orientation (GSO) (β = −0.296; *p* = 0.003), and no domination by symptoms NDS (β = −0.214; *p* = 0.025), both of which were statistically significant. This suggests that the older the patients, the lower their degree of recovery in relation to their hope for the future and the lower the desire to succeed; this is where psychiatric symptoms were no longer the focus of personal life.

All multiple linear regression models were statistically significant (PCH: F = 3.975, *p* < 0.001; WAH: F = 2.912, *p* = 0.001; GSO: F = 2.602, *p* = 0.002; RO: F = 2.318, *p* = 0.006; NDS: F = 3.726, *p* < 0.001) and able to predict between 13% and 24.3% of the variability in the levels of the recovery variable dimensions. Details of the multiple linear regression are presented in Table 3.

## 4. Discussion

The results of this study showed an association between medication adherence and personal recovery, especially the recovery of patients with schizophrenia. It should be noted that the adherence levels of these patients were classified as moderate to high (84.7%), which could influence the level of total recovery. Similarly, 25% of the patients presented an advanced level of recovery (mean scores above the 75th percentile [Q3 = 4.21]), wherein those who were young, had a partner, and a higher educational level, would show higher rates of personal recovery.

More specifically, adherence has a significant effect on the Willingness to Ask for Help dimension, which emphasizes the importance of incorporating and favoring spaces for interaction and communication between patients and mental health staff.

Previous studies have presented findings along the same lines of this study, highlighting that adherence to treatment is a fundamental aspect for optimizing recovery [61,62,63]. Thus, early and systematic symptom control in the course of patients’ disorder is an essential factor in improving their integration back into the community, with the latter being one of the most important elements of recovery [61,62,63,64,65].

The results also show a relationship between the symptomatological severity of the patient and his or her recovery; thus, negative symptoms present a significant relationship with the dimensions of Personal Confidence and Hope and Goal, and success orientation. Previous studies have shown that both positive and negative symptoms demonstrate a small to medium association with personal recovery and hope [65]; however, other studies have emphasized how negative symptoms are strongly associated with personal recovery, more so than positive symptoms [66,67].

Similarly, in this research, cognitive symptoms presented a significant relationship with Willingness to ask for help and no domination by symptoms dimensions of recovery. Similar to previous studies, cognitive symptoms are part of the category of disorder-related variables that influence recovery [68].

The findings of this study seem to show that the equation: the greater the adherence, the greater the remission of symptoms, and therefore, the greater the likelihood of recovery makes sense; however, in this study the latter is considered as a non-uniform, more long-term process that requires the interaction of more variables [65,66,69] that are also adequately incorporated into the treatment of patients and their families. Thus, the delivery and maintenance of pharmacological treatment could ensure periods of stability that facilitate the introduction of psychosocial interventions that combine psychoeducation and cognitive and behavioral management techniques, which together could promote the recovery of these patients [69,70,71,72,73,74]. 

Despite the above, it is necessary to consider that the recovery process involves a unique follow-up of each patient. At some point, if the patient achieves a degree of adequate recovery, they have the legitimate right to ask for a gradual reduction of their drug therapy. It has to be taken into account that prolonged treatment with antipsychotics can accrue detrimental effects on the physical health and brain structure of such patients [34,75].

The study has limitations: First, the cross-sectional design used does not allow the establishment of causal relationships. Second, the measurement of adherence to medication was based on a subjective assessment and did not include more objective measurements. Third, most of the patients were stable, which could imply some degree of selection bias and the possibility of choosing patients with better adherence. (They were recruited from a clinic check-up.) Fourth, the study did not include a classification of the type of antipsychotic medication used by the patients. Future research should consider the differential effects of classical and atypical antipsychotics, as well as incorporate other elements related to adherence, such as therapeutic alliance, time of untreated psychosis, and family and social support. Finally, the study considered only one area of recovery, personal recovery, which limits the generalization of the results to other areas of recovery.

Future research should consider a prospective longitudinal design that incorporates the measurement of other areas of recovery.

## 5. Conclusions

The greatest strength of this study is its analysis of the relationship between adherence and recovery, wherein a quantitative analysis design was used with standardized measures of the variables studied, achieving one of the first inputs to strengthen the treatment of these patients. This has not been previously studied in Latin American countries. 

Adherence has a positive relationship with personal recovery in this sample of Chilean patients with schizophrenia. An important effect of adherence was observed, especially in the Willingness to Ask for Help dimension of recovery, highlighting once again the need to maintain constant communication and alliance among the patient, their family, and the mental health services.

## Figures and Tables

**Table 1 healthcare-09-01230-t001:** Sociodemographic characteristics.

Patients		M (SD) ± Range or n (%)
Gender	Men	94 (62.3%)
	Women	57 (37.7%)
Age		39.9 (14.5) ± 14–75
Aymara	Yes	67 (44.3%)
	No	82 (54.3%)
	Missing	2 (1.4%)
Marital status	With a partner	30 (19.8%)
	Without a partner	121 (80.2%)
Educational Level	>12 years	14 (9.2%)
	<12 years	136 (90.0%)
	Missing	1 (0.8%)
Employment status	With employment	21 (13.9%)
	Without employment	128 (84.7%)
	Missing	2 (1.4%)
Religion	With a religion	110 (72.8%)
	Without a religion	38 (25.2%)
	Missing	3 (2.0%)
Family income (US dollars)		216.4 (206.3) 0–1820.1
Age of disease onset		21.4 (8.4) ± 8–50
Age of onset of treatment		23.8 (8.9) ± 11–50
Pharmacological treatment	Yes	151 (100%)
	No	0 (0%)
DAI-10 categorized	Non-adherent	23 (15.2%)
	Moderate	61 (40.3%)
	Adherent	67 (44.4%)
RAS-24 total		3.7 (0.7) ± 1.6–4.9
PANSS categorized	Mildly ill	64 (42.4%)
	Moderately ill	55 (36.34%)
	Markedly ill	25 (16.6%)
	Severely ill	7 (4.6%)
PANSS total		61.1 (18.6) ± 30.0–111.0

Note: M = mean; SD = standard deviation; n = number of subjects; % = effective (percentage); DAI-10 = Drug Attitude Inventory; RAS = Recovery Assessment Scale; PANSS = Positive and Negative Syndrome Scale; LA-ISMI = The Latin American version of internalized stigma of mental illness scale.

**Table 2 healthcare-09-01230-t002:** Descriptive analysis of quantitative variables and correlation matrix between study variables.

Variable	M (SD)	Min-Max	S	K	Shapiro-Wilk	2	3	4	5	6	7	8	9	10	11
1. DAI-10	3.7 (4.2)	−10.0–10.0	−3.2	−0.4	**0.938 ***	0.155	**0.284 ***	0.085	**0.184**	**0.187**	−0.016	−0.090	−0.087	−0.073	**−0.164**
2. PCH	3.6 (0.7)	1.2–5.0	−2.9	−0.5	**0.964 ***		**0.541 ***	**0.610 ***	**0.428 ***	**0.507 ***	**−0.255**	**−0.438 ***	**−0.356 ***	**−0.205**	**−0.344 ***
3. WAH	3.8 (1.1)	1.0–5.0	−4.5	0.1	**0.866 ***			**0.444 ***	**0.515 ***	**0.425 ***	−0.064	**−0.198**	**−0.306 ***	0.047	**−0.183**
4. GSO	3.7 (1.0)	1.0–5.0	−5.1	1.6	**0.900 ***				**0.297 ***	**0.355 ***	−0.138	**−0.305 ***	**−0.245**	−0.047	**−0.217**
5. RO	3.8 (0.8)	1.0–5.0	−3.7	0.9	**0.940 ***					**0.344 ***	−0.049	**−0.188**	**−0.249**	0.038	**−0.162**
6. NDS	3.4 (1.0)	1.0–5.0	−2.4	−0.7	**0.941 ***						−0.120	**−0.322 ***	**−0.412 ***	−0.088	**−0.247**
7. PANSS positive	10.2 (3.9)	5.0–25.0	4.1	1.5	**0.940 ***							**0.405 ***	**0.514 ***	**0.494 ***	**0.609 ***
8. PANSS negative	15.9 (6.2)	7.0–37.0	3.0	0.1	**0.959 ***								**0.623 ***	**0.319 ***	**0.421 ***
9. PANSS cognitive	6.0 (2.6)	3.0–15.0	4.7	1.8	**0.910 ***									**0.291 ***	**0.597 ***
10. PANSS depression	7.9 (3.1)	4.0–18.0	2.9	−0.3	**0.935 ***										**0.588 ***
11. PANSS excitement	8.7 (3.4)	5.0–19.0	3.8	0.0	**0.905 ***										

Note: M = mean; SD = standard deviation; Min-Max = minimum and maximum; Sa = standardized skewness; K = standardized kurtosis; Shapiro-Wilk = Shapiro-Wilk test; Values in bold indicate a statistically significant (*p* < 0.05); Values in bold and asterisk (*) indicate statistical significance (*p* < 0.001); PCH = Personal confidence and hope; WAH = Willingness to ask for help; GSO = Goal and success orientation; RO = Reliance on others; NDS = No domination by symptoms; DAI−10 = Drug Attitude Inventory; PANSS = Positive and Negative Syndrome Scale.

**Table 3 healthcare-09-01230-t003:** Results of Multiple Regression Analysis by Dimension of RAS.

Dimensions	PCH		WAH		GSO		RO		NDS	
F_(df)_	3.975_(15)_		2.912_(15)_		2.602_(15)_		2.318_(15)_		3726_(15)_	
*p*	<0.001		0.001		0.002		0.006		<0.001	
Adj. R^2^	0.243		0.171		0.147		0.130		0.227	
Standardized Coefficient & *p*-value										
	β	*p*	β	*p*	β	*p*	β	*p*	β	*p*
Intercept	–	–	–	–	–	–	–	–	–	–
Gender	0.016	0.843	0.084	0.334	−0.010	0.907	−0.027	0.764	0.008	0.923
Marital Status	0.032	0.700	−0.088	0.315	−0.107	0.225	−0.181	**0.045**	0.082	0.329
Age	−0.217	**0.021**	−0.032	0.742	−0.296	**0.003**	0.007	0.946	−0.214	**0.025**
Aymara	0.041	0.605	−0.027	0.739	0.065	0.439	0.088	0.299	0.072	0.363
Religion	0.056	0.504	0.009	0.919	0.027	0.763	0.022	0.805	0.072	0.398
Employment status	0.091	0.257	−0.062	0.461	0.035	0.678	0.031	0.718	0.086	0.289
Educational Level	0.106	0.166	0.010	0.896	0.104	0.201	0.000	0.998	0.086	0.267
Age of disease onset	0.177	0.250	−0.061	0.702	−0.062	0.703	0.094	0.570	0.183	0.238
Age of onset of treatment	−0.020	0.903	0.138	0.416	0.170	0.321	−0.241	0.166	0.003	0.984
DAI-10	0.096	0.226	0.239	**0.005**	0.035	0.679	0.144	0.092	0.142	0.077
PANSS positive	−0.023	0.825	−0.019	0.861	0.063	0.574	0.095	0.403	0.135	0.207
PANSS negative	−0.341	**0.001**	−0.082	0.441	−0.266	**0.014**	−0.175	0.109	−0.155	0.131
PANSS cognitive	−0.035	0.774	−0.305	**0.018**	−0.021	0.869	−0.119	0.365	−0.351	**0.005**
PANSS depression	−0.104	0.428	0.002	0.990	−0.124	0.372	−0.176	0.211	−0.065	0.620
PANSS excitement	−0.008	0.929	0.127	0.200	0.062	0.539	0.177	0.084	0.049	0.609

Note: F = Statistical F; *p* = Significance; adj. R^2^ = Coefficient R squared corrected; β = Standardized regression coefficient; Values in bold indicate a statistically significant (*p* < 0.05); PCH = Personal Confidence and Hope; WAH = Willingness to Ask for Help; GSO = Goal and Success Orientation; RO = Reliance on Others; NDS = No Domination by Symptoms; DAI-10 = Drug Attitude Inventory; PANSS = Positive and Negative Syndrome Scale.

## Data Availability

The data presented in this study are available on request from the corresponding author.

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
