# Peer review of "Effects of Adherence to Pharmacological Treatment on the Recovery of Patients with Schizophrenia"

_healthcare, 2021, doi:10.3390/healthcare9091230_

Round 1

Reviewer 1 Report

This study is confirming the previous finding that adherence to pharmacological treatment leads to better outcome and recovery for schizophrenia patients. However, the unique aspect of this study is the Chilean patients who are included in the study. The main weakness of the study which is also mentioned by the authors is that most majority of the patients are stable and adhering schizophrenia patients who are expected to experience a better recovery with medication.

Author Response

We appreciate the review carried out. We have sent the paper for English editing to a specialized company so this point was improved. Certificate in attachment.

Reviewer 2 Report

The article is interesting, the topic is actual. But, the patient adherence to treatment, as can be assumed, is, firstly, important not only in the treatment of schizophrenia, and secondly, of course, it is important for recovery. This topic and this conclusion cannot be called very original.

More than 50% of the references used are not included in the 5-year interval 2016-2020. I am sure that there is a sufficient amount of modern literature on this issue, the use of which, perhaps, would change the content of this article.

The article is well written, read with interest.

Author Response

We are grateful for the reviewer's comments, we have taken his/her suggestion into account and have incorporated an updated bibliography:

Sriramulu, S. B., Elangovan, A. R., Isaac, M., & Kalyanasundaram, J. R. Treatment non-adherence pattern among persons with neuropsychiatric disorders: A study from a rural community mental health centre in India. International Journal of Social Psychiatry 2021, Published. https://doi.org/10.1177/00207640211008462.

Viveiros, C. P., Tatar, C. R., dos Santos, D. V. D., Stefanello, S., & Nisihara, R. (2020). Evaluation of nonadherence to treatment among patients with schizophrenia attending psychosocial care centers in the south region of Brazil. Trends in Psychiatry and Psychotherapy, 42(3), 223–229. https://doi.org/10.1590/2237-6089-2019-0072

Blum, G. B., Bins, R. B., Rabelo-da-Ponte, F. D., & Passos, I. C. (2021). Stigmatizing attitudes toward psychiatric disorders: A cross-sectional population-based survey. Journal of Psychiatric Research, 139, 179–184. doi:10.1016/j.jpsychires.2021.05.033

Pu, C., Huang, B., Zhou, T., Cheng, Z., Wang, Y., Shi, C., & Yu, X. (2020). Gender Differences in the First-Year Antipsychotic Treatment for Chinese First-Episode Schizophrenia. Neuropsychiatric Disease and Treatment, Volume 16, 3145–3152. https://doi.org/10.2147/ndt.s280719

Johansen, K. K., Hounsgaard, L., Frandsen, T. F., Fluttert, F. A. J., & Hansen, J. P. (2020). Relapse prevention in ambulant mental health care tailored to patients with schizophrenia or bipolar disorder. Journal of Psychiatric and Mental Health Nursing. Published. https://doi.org/10.1111/jpm.12716

Joung, J., & Kim, S. (2017). Effects of a Relapse Prevention Program on Insight, Empowerment and Treatment Adherence in Patients with Schizophrenia. Journal of Korean Academy of Nursing, 47(2), 188. doi:10.4040/jkan.2017.47.2.188

Reviewer 3 Report

In this manuscript is presented a study on the Effects of Adherence to Pharmacological Treatment on the Recovery of Patients with Schizophrenia and the obtained results. The topic is relevant for clinical care; however, it is a local study and in my point of view only is more relevant for the Latin American population. Additionally, the text needs to be worked, because some phrases are very confusing and longer. An intensely English review is needed. I recommend a review of the text by a native English.

I would like to suggest several modifications to improve the text

Some examples of phrases with confuse ideas:

Line 37: “Difficulties with adherence in Schizophrenia, generates relapses in patients, which on average can be five times more in young men, in their first psychotic episode, than in women, increasing the likelihood of hospitalizations and healthcare costs [4-6].”

Line 256: “This suggests that the older the age of the patients, the lower the degree of recovery in relation to hope for the future, desires to succeed, and psychiatric symptoms no longer being the focus of personal life.”

Line: 307: “Despite the above, it is necessary to consider that the recovery process involves an individualized follow-up of each patient, given that at some point, if the patient achieves  an adequate recovery, he/she has the legitimate right to opt for a gradual reduction of  his/her pharmacological therapy, also considering that prolonged treatment with antipsychotics can generate accumulated detrimental effects on the physical health and brain  structure of these patients [29, 69]”

These are only some examples, an extensively review of all text its need.

In addition, I have other major points to indicate:

Line 34- “(p.1)”. Eliminated this, it is no needed here.

Line 34-“The World Health Organization (WHO) considers adherence to treatment as adherence to treatment; that is, taking medication according to the prescribed dosage and persistence, over time [3]”.  This phrase is very confuse, please revise it.

Line 80- a explanation about the definition of the Drug Attitude Inventory (DAI-10), Recovery Assessment Scale (RAS-24) and Positive and Negative Syndrome Scale for Schizophrenia (PANSS) should be in the introduction. Please review these points in the text.

Table 1:

Pharmacological treatment

Yes

151 (100%)

                                                  No

                          0 (0%)

                                                  No

                          121 (80.1%)

Please explain why in the table you wrote twice “no”

Line 199- “(M = 3.7, SD = 4.2) “, in the table the authors wrote SD 0.7. The authors should clarify the point.

Author Response

We appreciate the review carried out. We have corrected the points suggested by the reviewer and also We have sent the paper for English editing to a specialized company (certificate in attachment).

Reviewer 4 Report

The manuscript by Caqueo-Urízar et al., is a retrospective study about the effects of adherence to antipsychotic treatment on patients with schizophrenia in northern Chile. The study includes the analysis of several parameters obtained by 151 patients.

Overall, this research is well written, the tables are useful, and the content of this manuscript is of major interest. I do not find any significant incorrectness. My following comments are of minor character:

Line 20: “criterio variable” is Spanish. Maybe the authors want to say “variable criterion”.

Line 34: Please delete the typo “(p.1)”

Lines 37-39: This sentence should be rephrased. For instance, you can write: ...generates, in their first psychotic episode, relapses which on average...

Line 53: Please avoid repetitions. Here “However” can be removed.

Line 68: Study participants

Line 352: Please check the reference list style. Besides, some references need corrections

Author Response

(The authors gave the same response as above.)

Round 2

Reviewer 3 Report

The authors clearly improved the manuscript, therefore, in my opinion, it is now susceptible for publication